

# Evaluating ocean alkalinity enhancement as a carbon dioxide removal strategy in the North Sea

Feifei Liu[1], Ute Daewel[1], Jan Kossack[1], Kubilay Timur Demir[1], Helmuth Thomas[2], Corinna Schrum[1,3]

[1]Institute of Coastal Systems, Helmholtz-Zentrum Hereon, Geesthacht, Germany
[2]Institute of Carbon Cycles, Helmholtz-Zentrum Hereon, Geesthacht, Germany
[3]Institute of Oceanography, University of Hamburg, Hamburg, Germany

*Correspondence to*: Feifei Liu (feifei.liu@hereon.de)

**Abstract.** Ocean Alkalinity Enhancement (OAE) is a climate mitigation strategy aimed at increasing the ocean's capacity to absorb and store atmospheric $CO_2$. The effect of OAE depends significantly on local physical conditions, underscoring the importance of selecting optimal locations for alkalinity addition. Using a regional coupled physical-biogeochemical-carbon model, we examine OAE responses in the North Sea, including $CO_2$ uptake potential, enhanced carbon storage and cross-shelf export, and the associated changes in the carbonate chemistry. Alkalinity is continuously added as a surface flux in three
distinct regions of the North Sea. Our simulations show that the Norwegian Trench and the Skagerrak serve as sinks for added alkalinity, reducing its interaction with the atmosphere. Alkalinity addition along shallow eastern coasts results in a higher $CO_2$ uptake efficiency (~0.79 mol $CO_2$ uptake per mol alkalinity addition) than offshore addition in ship-accessible areas (~0.66 mol $CO_2$ uptake per mol alkalinity addition), as offshore alkalinity is more susceptible to deep-ocean loss. Long-term carbon storage, measured by excess carbon accumulation in deep ocean layers and cross-shelf export below permanent
pycnoclines, is similar across the three scenarios, accounting for less than 10% of total excess $CO_2$ uptake. The smallest changes in pH occur when alkalinity is added offshore, with effects nearly an order of magnitude lower than alkalinity addition in the shallow German Exclusive Economic Zone, where pH increases from 8.1 to 8.4. The model's resolution (~4.5 km in coastal areas) limits its ability to capture rapid, localized carbonate responses, leading to a nearly tenfold underestimation of chemical perturbations. Thus, finer-scale models are needed to accurately assess near-source alkalinity impacts.

## 1 Introduction

Ocean alkalinity enhancement (OAE) is a proposed geoengineering strategy aimed at accelerating the uptake of atmospheric carbon dioxide ($CO_2$) by manipulating marine carbonate chemistry through increasing the alkalinity of seawater (Renforth and Henderson, 2017). As a potential approach for carbon dioxide removal (CDR), OAE has recently garnered significant interest,



driven by the substantial carbon storage potential of Earth's oceans and the flexibility of the strategy to be implemented. The rationale behind OAE is to leverage the marine carbonate system (Renforth and Henderson, 2017), a multiple equilibrium state (Zeebe and Wolf-Gladrow, 2001) described by the reaction:

$$CO_{2(aq)} + H_2O \rightleftharpoons H^+ + HCO_3^- \rightleftharpoons 2H^+ + CO_3^{2-}$$    Eq.(1)

Dissolved inorganic carbon (DIC) is the combined concentration of all carbonate species. Addition of alkalinity (e.g., $OH^-$)
shifts the above equilibrium to the right by consuming $H^+$ ions and allows $CO_2$ to be stored in the ocean as abundant and stable carbonate and bicarbonate ions. Accordingly, the partial pressure of $CO_2$ ($pCO_2$) in the ocean is lowered, which drives further oceanic $CO_2$ uptake (Zeebe and Wolf-Gladrow, 2001) and might also mitigate ocean acidification (Bach et al., 2019; Hartmann et al., 2013). OAE approaches include accelerating mineral weathering by adding finely ground rocks to corrosive or high-weathering environments (Foteinis et al., 2023; Hangx and Spiers, 2009; Montserrat et al., 2017; Rigopoulos et al., 2018);
adding more rapidly dissolving substances, such as quick lime/lime ($CaO$, $Ca(OH)_2$) to the ocean (Kheshgi, 1995; Paquay and Zeebe, 2013; Renforth and Kruger, 2013); and electrochemical methods which produce highly alkaline solutions that can be discharged to the ocean (Davies et al., 2018; Digdaya et al., 2020).

Coastal seas, of high efficiency in exporting carbon to the open ocean, contribute significantly to the global oceanic uptake of atmospheric $CO_2$ (Frankignoulle and Borges, 2001; Laruelle et al., 2014; Legge et al., 2020). Due to their easy accessibility,
relatively shallow depth and the proximity to mineral and energy sources, they present promising opportunities for OAE implementations through a practical and economic lens (Hangx and Spiers, 2009; He and Tyka, 2023). Coastal regions are easily accessible for the deployment of infrastructure and equipment, as well as the transport of alkaline substances and vessels or platforms for their distribution (Foteinis et al., 2022). The OAE deployments could also be integrated into existing coastal zone management practices such as dredging operations, land reclamation and beach nourishments (Montserrat et al., 2017).
The highly dynamic coastal environments, induced by tidal movements and wave action, facilitate the dissolution and reaction of the alkaline materials with $CO_2$.

Major concerns regarding coastal OAE applications include the efficiency of $CO_2$ uptake, potential side effects on ecosystems due to drastic changes in oceanic conditions (Bach et al., 2019), and the risk of secondary mineral precipitation (Hartmann et al., 2023; Moras et al., 2022). These concerns are largely influenced by regional-scale oceanic conditions, including circulation
patterns and mixing processes, which exert strong control on the distribution and effectiveness of added alkaline materials. Ocean currents help disperse these materials, promoting widespread alkalinity changes rather than localized effects. Vertical mixing can lead to a loss of alkalinity to deeper waters, potentially lowering $CO_2$ uptake efficiency. The lost alkalinity might remix into the upper mixed layer at some later time, driving further $CO_2$ uptake elsewhere and over longer time scales (He and Tyka, 2023). Coastal areas are often heavily impacted by pollution and eutrophication from human activities such as urban





runoff, industrial discharge, and agricultural runoff. Adding alkaline substances to these waters could exacerbate existing ecological problems by introducing co-contaminants, such as iron, silicate or heavy metals, into the ocean (Bach et al., 2019; Ferderer et al., 2022; Guo et al., 2022; Zhu et al., 2024). Uneven distribution of alkalinity enhancement due to mixing and circulation could lead to localized elevation in pH and other water chemistry parameters, causing unintended consequences for marine ecosystems (Bach et al., 2019; Subhas et al., 2022). Furthermore, increases in aragonite saturation, caused by the

increase of alkalinity, could lead to the precipitation of calcium carbonate, which removes alkalinity from the surface water and is counterproductive to $CO_2$ uptake (Hartmann et al., 2023; Moras et al., 2022). Careful consideration of these associated challenges and risks is necessary to ensure the effectiveness and sustainability of the coastal OAE practices.

Several previous studies have incorporated physical effects into their hypothetical OAE scenarios by using ocean circulation models, most of which have focused on the global scale. These studies primarily examine open oceans (e.g. Burt et al., 2021;

González and Ilyina, 2016; Ilyina et al., 2013; Keller et al., 2014; Köhler et al., 2013) or large patches of the coastal surface ocean (e.g. Feng et al., 2017; Hangx and Spiers, 2009; He and Tyka, 2023; Palmiéri and Yool, 2024). Using global simulations, Burt et al. (2021) found that physical regimes, along with alkalinity sensitivity, play key roles in driving the carbon uptake response to different regional OAE implementations. Similarly, He and Tyka (2023) demonstrated that the efficiency of $CO_2$ uptake varies geographically by simulating pulse alkalinity injections at various coastal locations. Their findings attribute these

variations to differences in equilibration kinetics, which are influenced by the interplay between local gas exchange rates and surface water residence time. However, regional-scale studies on OAE in the context of operational practices remain relatively scarce, despite the clear advantages of regional models in capturing coastal and continental shelf processes (Laurent et al., 2021). For example, Butenschön et al. (2021) employed a coupled regional physical-biogeochemical model for the Mediterranean Sea to investigate the efficient removal of atmospheric $CO_2$ and the mitigation of ocean acidification. Wang et

al. (2023) used a coupled regional ice–circulation–biogeochemical model to evaluate the efficiency and ocean acidification mitigation impacts of a sustained point-source OAE in the Bering Sea.

In this paper, we focus our modelling study on OAE scenarios in the North Sea, a coastal region adjacent to the North Atlantic with a mean depth of 80 meters and a maximum depth of 800 meters in the Norwegian Trench. The physical structure of the North Sea is shaped by tidal forcing, wind effects, and topography, resulting in distinct hydrodynamic regimes: shallow

permanently mixed areas, transitional regions with tidal fronts, seasonal stratified regions, and deep stably stratified regions (Van Leeuwen et al., 2015). The interaction of tides, meridional density gradient, and wind patterns creates heterogeneous flushing dynamics across the North Sea, resulting in spatial variability in water residence time (Blaas et al., 2001), which is a key factor in determining OAE efficiency (He and Tyka, 2023). This physical complexity underscores the importance of selecting suitable geographic locations for OAE. An optimal location would ensure that seawater with elevated alkalinity is

efficiently transported, allowing ample atmospheric contact before moving into the ocean's interior, thereby maximizing OAE efficiency while minimizing potential impacts on the local carbonate system and ecology. As a vital part of the global ocean



carbon sink, the North Sea efficiently exports carbon-enriched subsurface waters into the deep North Atlantic through the continental shelf pumping mechanism (Thomas et al., 2004), where cross-shelf exchange plays a key role (Graham et al., 2018; Guihou et al., 2018; Holt et al., 2009; Thomas et al., 2005, 2004). This raises an important question: to what extent might OAE

alter this lateral off-shelf carbon transport, and how might the specific locations of OAE deployment influence this effect?

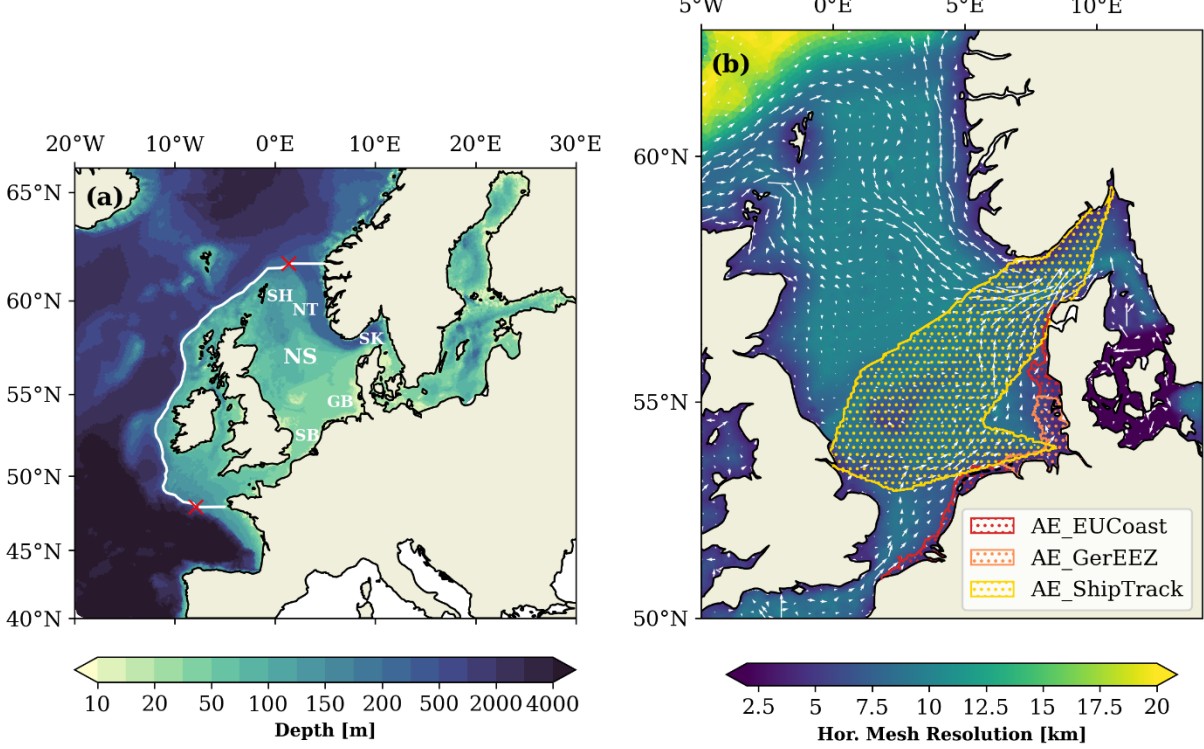

**Figure 1: (a) Map of the model domain with the bathymetry. White line represents location of the shelf break, defined as 200-meter isobath, with gates at the northern and southern limits. Red crosses represent the starting points of the gates. SH: Shetland, NT: Norwegian Trench, SK: Skagerrak, NS: North Sea, GB: German Bight, SB: Southern Bight. (b) Map of the model horizontal**
**resolution overlayed by the average current field over the period of 2001-2010 from the model simulation. The map is constrained to the North Sea for clarity. Hatched areas are OAE deployment sites in the three scenarios.**

To address those above questions, we simulate OAE scenarios by assuming that pure alkalinity is added to the surface waters at selected locations. We do not address additional constraints related to practical OAE implementations, such as mineral weathering. As an initial step in OAE assessment, our focus is on the impact of alkalinity addition beyond the immediate

application sites, where regional-scale processes predominantly influence the response of the North Sea system, rather than the specific methods of alkalinity introduction.

We use a three-dimensional coupled ocean-biogeochemical-carbon model to track the spatial and temporal distribution of added alkalinity and the resulting anomalies in air–sea $CO_2$ exchange. The model further enables us to estimate the





spatiotemporal extent of OAE-induced changes in other carbonate chemistry parameters. Our objectives are to: 1) trace the
fate of added alkalinity; 2) assess the efficiency of $CO_2$ uptake driven by OAE over multi-year timescales, including its storage
and export potential; and 3) quantify the spatial and temporal extent of changes in carbonate system properties. This approach
provides critical insights into the potential response of the North Sea system to real-world OAE implementations.

## 2 Methods

### 2.1 Model description

In this study, we apply the coupled physical-biogeochemical-carbon regional model framework SCHISM-ECOSMO-CO2
described in Kossack et al. (2024). The three dimensional hydrodynamic model SCHISM is coupled with the lower trophic
level ecosystem model ECOSMO II and the carbon module via the FABM framework (The Framework for Aquatic
Biogeochemical Models, Bruggeman and Bolding 2014).

The physical model SCHISM (Semi-implicit Cross-scale Hydroscience Integrated System Model) is used to provide three-
dimensional flow fields, turbulent mixing schemes, and other oceanic properties such as temperature and salinity to the
biogeochemical calculations. The model uses a highly flexible three dimensional gridding system, with a hybrid quadrangular-
triangular unstructured mesh in the horizontal dimension and localized sigma coordinates with shave cells (LSC$^2$) in the vertical
(Zhang et al., 2015). This innovation allows a seamless simulation of the three-dimensional baroclinic circulation across
various scales, ensuring that both the coastal and continental shelf processes are well represented (Ye et al., 2018; Yu et al.,
2017; Zhang et al., 2015; Zhang et al, 2016). The model captures key physical features on the North Sea, such as the regional-
scale circulation and the tide-induced spatial heterogeneity in mixing-stratification dynamics (Kossack et al., 2023), which
play an important role in redistributing the added alkalinity over the model domain.

The ecosystem model (ECOSMO II) simulates the oceanic cycles of carbon and other biogeochemical elements through a
nutrient-phytoplankton-zooplankton-detritus (NPZD) conceptual model approach, which encompasses three macronutrient
species (nitrogen, phosphorus and silicate), three phytoplankton functional groups (diatoms, flagellates and cyanobacteria),
two zooplankton functional groups (herbivorous and omnivorous) and three functional groups for detritus (particulate organic
matter, dissolved organic matter, biogenic opal) (Daewel and Schrum, 2013). Carbon fixation by autotrophs is restricted to the
elemental Redfield ratio (Redfield et al., 1963). The transport of tracers and organisms in space and time is described as a non-
linear advection and diffusion process provided by SCHISM. Benthic-pelagic coupling is realized through the introduction of
three single layer sediment pools, with one for opal, one for particulate organic material consisting of carbon and nitrogen and
one for iron-bounded phosphorous. The total alkalinity (TA) is prognostically calculated based on the contributions from nitric
and phosphoric acid systems following its explicit conservative expression explained by Wolf-gladrow et al. (2007), as is
expressed:



$$\frac{dTA}{dt} = \sum_{bio} \frac{d(NH_4)}{dt} - \sum_{bio} \frac{d(NO_3)}{dt} - \sum_{bio} \frac{d(PO_4)}{dt} \qquad \text{Eq.(2)}$$

No feedback from alkalinity change to biological processes is considered. Calcification or calcium carbonate precipitation is not included in the model.

Furthermore, a carbonate module originated from Blackford and Gilbert (2007) is embedded into the modelling system to resolve the inorganic carbon chemistry and air-sea gas exchange based on two prognostic tracers provided from ECOSMO: DIC and TA. Three carbon species ($CO_2$, $HCO_3^-$ and $CO_3^{2-}$) under the local temperature and salinity conditions are calculated

based on Millero et al. (2006). Other key parameters of the carbon system, such as $pCO_2$ and the sea water pH, are iteratively solved. The air-sea exchange is determined by the sea surface wind speed and the difference of $pCO_2$ in the sea surface and the air above through the gas transfer parameterization of Wanninkhof (2014).

Within the SCHISM-ECOSMO-CO2 system, interactions between the ocean and ecosystem components and the carbonate system are unidirectional. There are no direct or indirect feedback from the carbonate system on any physical or ecosystem

state variables. Consequently, differences between the reference and OAE simulations presented below can be attributed solely to the passive advection and diffusion of the elevated alkalinity. While potential biogeochemical feedback may occur in the North Sea, incorporating this added complexity is beyond the scope of this study. Here, we aim to isolate and simulate the direct effects of alkalinity increase as a first-order approach.

**2.2 Model configuration**

The model domain encompasses the entire Northwest European Shelf (NWES), the Baltic Sea, and part of the northeast Atlantic, spanning 40°N– 66°N and 20°W – 30°E. We extend the model domain beyond the North Sea to this larger area because the anomalies in air–sea $CO_2$ flux generated by OAE deployments might manifest over extensive spatiotemporal scales, as the gas exchange between the ocean and the atmosphere is quite slow. Furthermore, the large domain including the North Atlantic here allows a realistic simulation of cross-shelf exchange (Kossack et al. 2023).

Further details on the model configuration, including thorough validation of the physical dynamics, biological processes (e.g. primary production) and carbonate chemistry, are provided in Kossack et al. (2023; 2024). We use the same configuration but at a coarser resolution, which ranges from approximately 4.5 km in shallow coastal areas to around 15 km in the North Atlantic. Vertically, the model achieves high resolution, with 2.5–6 m layer spacing from the surface to approximately 60-meter depth to capture the regional thermocline in detail. The number of vertical layers varies with water depth, from 3 layers in shallow

waters (minimum depth of 10 meters) to 52 layers in areas deeper than 4000 meters.



At the sea surface, atmospheric forcings are provided by a hindcast simulation with the regional atmospheric model COSMO-CLM, which has a horizontal resolution of 0.11° at an hourly time step (Geyer, 2017). We include a domain-wise correction to the shortwave radiation by +15% to account for a domain-wide sea surface temperature (SST) cooling bias (Kossack et al., 2023). The net shortwave radiation is calculated from a constant albedo of 0.06 while the upward longwave radiation is calculated from the modelled SST. The penetration of shortwave radiation into the ocean is calculated according to Jerlov optical water type IA in the whole model domain (Jerlov 1976).

The simulation is initialized from WOA2018 for the pelagic fields of temperature, salinity, nutrients (nitrate, phosphate, silicate) and oxygen (Boyer et al., 2018), while initial conditions for DIC and alkalinity are derived from NNGv2LDEO climatology (Broullón et al., 2019, 2020), except for the Baltic, where alkalinity initial conditions are generated using the end-member approach following Hjalmarsson et al. (2008). The sediment fields are initialized from long-term ECOSMO simulations provided by Samuelsen et al. (2022).

At the open boundaries in the North Atlantic Ocean, a sponge zone is set, over which the modelled temperature, salinity, nutrients (nitrate, phosphate, silicate) and oxygen are relaxed towards climatological monthly fields from the World Ocean Atlas (Boyer et al., 2018). The relaxing of DIC and alkalinity considers both the climatology from the global NNGv2LDEO climatology (Broullón et al., 2019, 2020) and the long-term trend from the monthly anomalies of the global ICON-Coast simulation (Mathis et al., 2022) to account for the effect of rising atmospheric $CO_2$. Atmospheric $pCO_2$ is prescribed from monthly mean data measured at the Mace Head station (Lan et al., 2022)

River forcing is provided in form of daily discharge and nutrients, DIC and alkalinity loads for the top 150 largest rivers in the domain from Daewel and Schrum (2013). Inputs for the rest of the ECOSMO state variables are set to zero. For the main rivers on the European continental coast (Scheldt, Meuse, Rhine, Ems and Elbe), the climatological concentrations of riverine DIC and alkalinity are adapted from the dataset compiled by Pätsch and Lenhart (2008), while for the British rivers such as the Humber estuary, Wear, Twead, Great Ouse and Thames, the multi-year averaged DIC and alkalinity are provided by the compilation of Neal and Davies (2003). Due to the lack of observation for the remaining rivers, we prescribe an average DIC concentration of 2700 mmolC/m3 computed from Pätsch and Lenhart (2008) and use the end member approach following Hjalmarsson et al., (2008) and Artioli et al. (2012) to estimate the alkalinity concentrations.

The model runs from 1995 to 2010 for the reference simulation (referred to as CTL hereafter). We treat the initial six years as a spin-up period and do not include them in subsequent analysis. The OAE scenario simulations commence from 2001 of the reference simulation and extend until the end of 2010.



### 2.3 OAE experiments

The OAE scenario experiments are identical to the reference run, with the only difference being the addition of a surface alkalinity flux to simulate OAE implementation. We set up three OAE scenarios, each based on a distinct area where alkalinity is added to the sea surface. AE_EUCoast is a scenario where the OAE area encompasses the European coastal bands, following shorelines with water depths of less than 20 meters. This area is considered economically viable, as facilities for producing alkaline material would likely be located on the coast, powered by the electrical grid, and use existing outfall pipes for dispersal.

In AE_GerEEZ, the OAE area is further restricted to the German Exclusive Economic Zone (EEZ), under the assumption that implementation is conducted solely by Germany. AE_ShipTrack involves distributing alkalinity into the ocean surface via cargo vessels that regularly transit between Immingham (UK), Cuxhaven (Germany) and Oslo (Norway) as part of the Coastal Observing System for Northern and Arctic Seas monitoring efforts (Baschek et al., 2017). In this scenario, we do not weight the alkalinity input by ship track density. This approach keeps the addition method consistent across the three scenarios,

ensuring that any differences arise solely from OAE locations, thus providing a first-order evaluation.

To simulate OAE, we add alkalinity as a continuous flux across the surface boundary at each of the three selected locations. In each scenario, the amount of added alkalinity is the same, targeting the removal of 5 megatons of $CO_2$ per year. Based on the typical $CO_2$ uptake efficiency of 0.85 moles of $CO_2$ removal per mole of added alkalinity (Montserrat et al., 2017), a total of 134 Gmol of alkalinity is added annually in the simulation. Due to differences in the sizes of the deployment sites, the rate

of alkalinity addition varies across scenarios (Table 1).

In all scenarios, we assume that the added alkalinity is from instantly dissolving substances such as sodium hydroxide ($NaOH$) solution. This assumption avoids complications associated with slower dissolving materials such as fine olivine powder, for which dissolution rates vary with ocean conditions and may sink out of the surface layers before complete dissolution (Fuhr et al., 2022).

An additional scenario (AE_EUCoast_Ter) is initialized from AE_EUCoast at the start of 2008 and runs until 2010. Over this period, no alkalinity is added to the ocean to simulate the termination of OAE. This scenario is only used in section 3.2 to evaluate $CO_2$ uptake efficiency.






**Table.1 Summary of the scenarios**

| Scenario name | Deployment sites (Fig.1) | Where the alkalinity is added | Addition frequency | Deployment period | Deployment area size (km²) | Amount added per unit (mmol/m²/s) |
|---|---|---|---|---|---|---|
| AE_EUCoast | European coast (<20m) | Surface | continuously | 2001-2010 | 28443 | 1.5e⁻⁴ |
| AE_EUCoast_Ter | - | - | - | 2008-2010 (Initialized from 2008-01 of AE_EUCoast) | 0 | 0 |
| AE_GerEEZ | German EEZ in the North Sea (<20m) | Surface | continuously | 2001-2010 | 11901 | 3.57e⁻⁴ |
| AE_ShipTrack | Area covered by regular ships | Surface | continuously | 2001-2010 | 184962 | 2.3e⁻⁵ |


## 2.4 Calculation of mixed layer depth and the permanent pycnocline

The mixed layer depth is used to divide the water column into two layers: the upper mixed layer where alkalinity remains in contact with the atmosphere and can realize its $CO_2$ uptake potential, and the layer below where alkalinity is isolated from the atmosphere and thus loses the opportunity to contribute to $CO_2$ uptake.

The mixed layer depth is calculated based on the modeled daily density field. The water column is classified as "stratified" if the density difference between the surface and bottom exceeds 0.086 kg/m³ (Van Leeuwen et al., 2015). Otherwise, it is considered "well mixed." For stratified waters, the mixed layer depth is defined as the depth of the maximum density gradient in the water column, following Carvalho et al. (2017). Permanent pycnocline only exists in areas where the water is stratified for more than 345 days per year (Van Leeuwen et al., 2015), and the depth of the permanent pycnocline is defined as the

maximum of the mixed layer depth over 2001-2010 (Holt et al., 2009).

## 2.5 Calculation of $CO_2$ uptake efficiency

The $CO_2$ uptake efficiency is conventionally defined as the ratio of cumulative DIC increase over the cumulative added alkalinity:



$$\eta CO_2(t) = \frac{\Delta DIC(t)}{\Delta ALK} \qquad\qquad\qquad\qquad\qquad\qquad\qquad\qquad\qquad\qquad\qquad Eq.(3)$$

where $\Delta ALK$ is the total alkalinity added and $\Delta DIC(t)$ is the excess total DIC inventory in the OAE scenarios relative to conditions without OAE at the time $t$ since alkalinity is added (Butenschön et al., 2021; He and Tyka, 2023; Renforth and Henderson, 2017).  Regional models lose tracers across lateral boundaries, making it unsuitable to use the proposed metric for estimating CDR efficiency. Instead, we use the temporal integral of enhanced $CO_2$ flux, which is also integrated spatially across the modelled domain, to account for the direct impact of $CO_2$ uptake. $\eta CO_2(t)$ is thus can be represented as:

$$\eta CO_2(t) = \frac{\int_0^t \Delta Flux(CO_2)}{\Delta ALK} \qquad\qquad\qquad\qquad\qquad\qquad\qquad\qquad\qquad\qquad Eq.(4)$$

This might result in a lower efficiency than theoretically expected, as some alkalinity is lost at the model boundaries and may still contribute to an additional $CO_2$ uptake outside the model domain. Additionally, because air-sea gas exchange responds more slowly than alkalinity addition, part of the alkalinity within the model domain has not yet fully realized its $CO_2$ uptake potential.

**3 Results**

**3.1 Alkalinity spreading**

We first examine the spreading of alkalinity, as this is crucial for determining whether OAE can achieve the desired effects over time. If the added alkalinity is mixed into the subsurface layer too quickly, the water mass will not have sufficient time to absorb atmospheric $CO_2$.

The evolution of excess alkalinity inventory across the entire model domain in all three scenarios shows a similar trend and magnitude over the simulation period. The excess inventory due to OAE increases nearly linearly during the first year, then slows in the second year, and by the third year, it reaches a plateau with minimal further growth (Fig. 2, solid lines). This suggests that the input of alkalinity is balanced by lateral transport out of the model domain, as strong water exchange prevents alkalinity buildup in the North Sea. The inventory exhibits distinct seasonality, peaking in summer and reaching its lowest
levels in winter.

Not all excess alkalinity in seawater effectively enhances the absorption of atmospheric $CO_2$, as part of the alkalinity is transported to the deep ocean, where it cannot equilibrate with the atmosphere until re-mixed to the surface. In the AE_EUCoast and AE_GerEEZ scenarios, approximately two-thirds of the total excess inventory remains in the upper mixed layer and supports effective atmospheric contact (Fig. 2, dashed lines). In contrast, the AE_ShipTrack scenario retains less alkalinity at



the surface, indicating a greater transport to the deep ocean. Surface excess alkalinity displays clear seasonality, peaking in

winter and out of phase with total excess alkalinity. This seasonal variability is most pronounced in the AE_ShipTrack scenario.

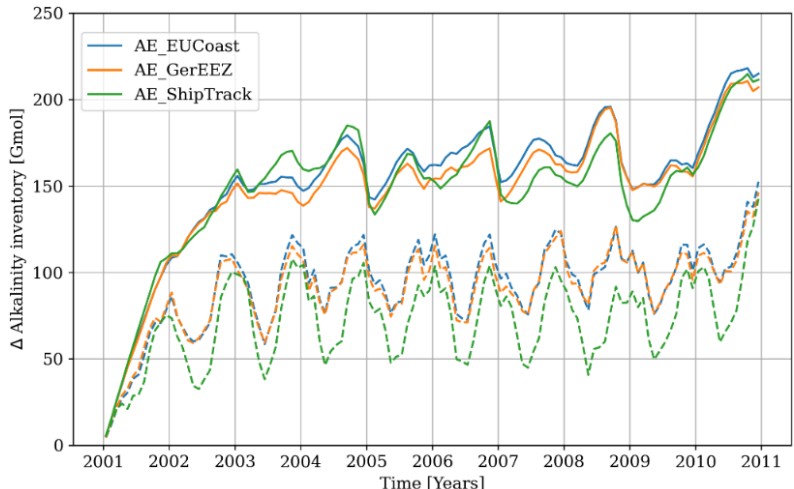

**Figure 2: The excess alkalinity inventory across the entire model domain, integrated over the entire water column (solid lines) and over the upper mixed layer (dashed lines).**

The spatial patterns of excess alkalinity, integrated above and below the mixed layer, provide insight into the mechanisms

distributing alkalinity both across the surface ocean and into deeper waters. In the upper mixed layer, the integrated alkalinity

anomalies relative to the CTL run exhibit similar patterns in the AE_EUCoast and AE_GerEEZ scenarios, with the highest

values observed along the German and Danish coasts (Figs. 3a and 3b). Offshore dispersal into the central North Sea occurs

due to tidal-induced advective fluxes (Rydberg et al., 1996), creating a gradient from coast to open sea. Differences are most

notable in the Southern and German Bights, directly resulting from the location of alkalinity addition. In the AE_ShipTrack

scenario, increased surface alkalinity is mostly confined to the deployment area before being transported to the Skagerrak,

with only limited dispersal toward the east coast (Fig. 3c). In all scenarios, horizontal alkalinity transport is strongly influenced

by prevailing currents. As a result, the excess alkalinity is transported northward by the anticlockwise coastal current to the

Skagerrak, from where it is further carried northward into the Norwegian Trench and the Atlantic. Overall, these patterns

highlight the variation in surface alkalinity retention and dispersal across the scenarios, with implications for the potential

atmospheric $CO_2$ uptake efficiency.

Below the mixed layer, a substantial portion of the excess alkalinity is retained within a narrow coastal band extending from

the Skagerrak to the Norwegian Trench (Figs. 3d-f). The AE_ShipTrack scenario exhibits a higher alkalinity inventory in this

region than the other coastal scenarios. In the AE_EUCoast and AE_GerEEZ scenarios, the transport pathway is clear: excess

alkalinity is horizontally transported from the addition sites to the Skagerrak, where the saline North Sea water mixes with the

fresher Baltic outflow. The prevailing cyclonic circulation further transports this water into the Norwegian Trench, during




which the enhanced alkalinity is gradually mixed into the deeper layers (Albretsen et al., 2012; Christensen et al., 2018). This process is further evidenced by the vertical alkalinity profile along a transect from the Norwegian coast to the Danish coast, as shown in Fig. 4. Results for the AE_GerEEZ scenario are not shown here, as they are similar with those of AE_EUCoast.

However, this transport regime alone does not fully explain the greater increase in deep-layer alkalinity observed in the AE_ShipTrack scenario (Fig.3f), especially given the lower surface inventory in the Skagerrak compared to the other scenarios (Fig.3c). Notably, the transect profile does not display the same vertical alkalinity gradient as seen in the AE_EUCoast scenario (Fig.4). Instead, a subsurface maximum increase is observed, suggesting that the deep alkalinity accumulation has an additional source beyond entrainment from above (Fig.4d).

Two passive tracer experiments (method described in Supplement S1 and Fig.S1) are conducted to illustrate the underlying mechanisms. In the first experiment, assuming that the surface Skagerrak is the only source region, the passive tracer is continuously added to this area. This setup does not result in a maximum subsurface increase (Figs. 5a-b and e). In contrast, continuous addition of the tracer to a selected surface area in the southern North Sea leads to a greater increase at subsurface depths (e.g. at 70 m depth) than at the surface along the Norwegian coast (Figs. 5c-d), resulting in a vertical profile similar to

the enhanced alkalinity observed in the AE_ShipTrack scenario (Fig. 5f). This suggests that local vertical transport in the southern/central North Sea supports a stimulated horizontal transport the of tracer below the mixed layer, leading to more efficient loss of the tracer to deeper layers.

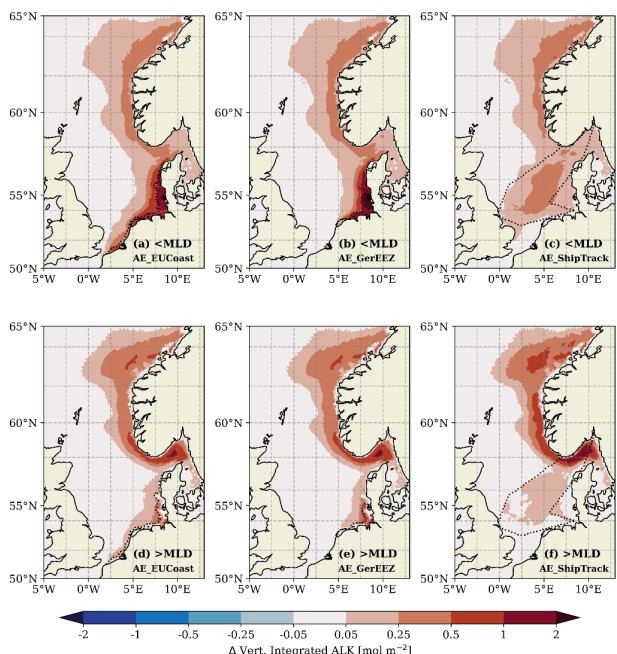

**Figure 3: Difference of vertically integrated alkalinity between the OAE scenarios and the CTL simulation over the upper mixed**
**layer (a-c) and below the mixed layer (d-f) in the last simulation year (2010). Note the nonlinear color bar.**



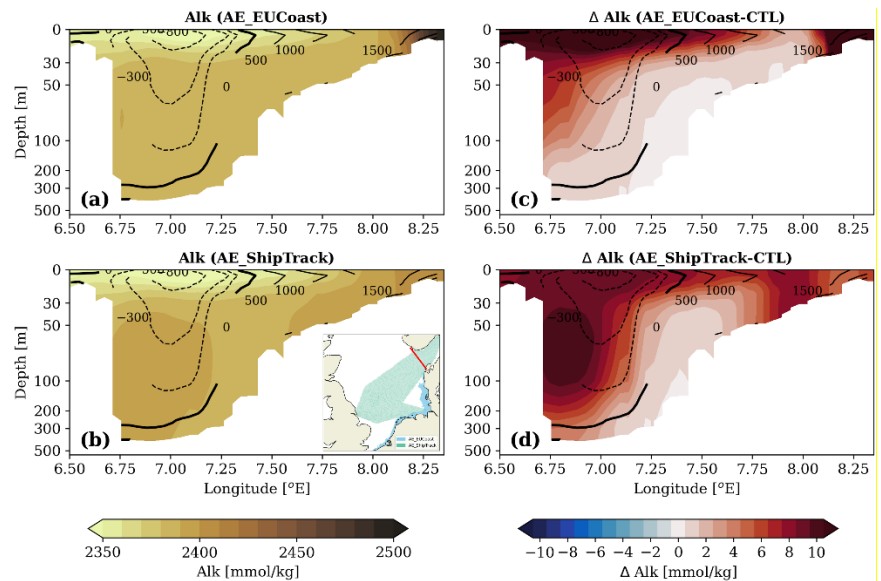

**Figure 4: (a)-(b)** Vertical distribution of the alkalinity concentration over the transect spanning from the Norwegian coast to the Danish coast (from left to right) for the AE_EUCoast and AE_ShipTrack scenarios in a typical month of February 2010. **(c)-(d)** The elevated alkalinity concentration due to OAE over this transect. Contour lines show the horizontal volume transport with the unit of m2/s, positive values represent inflow from the North Sea to the Skagerrak while negative values mean outflow from the Skagerrak to the Norwegian Trench. The location of the transect is highlighted by the red line in the inserted map in (b).

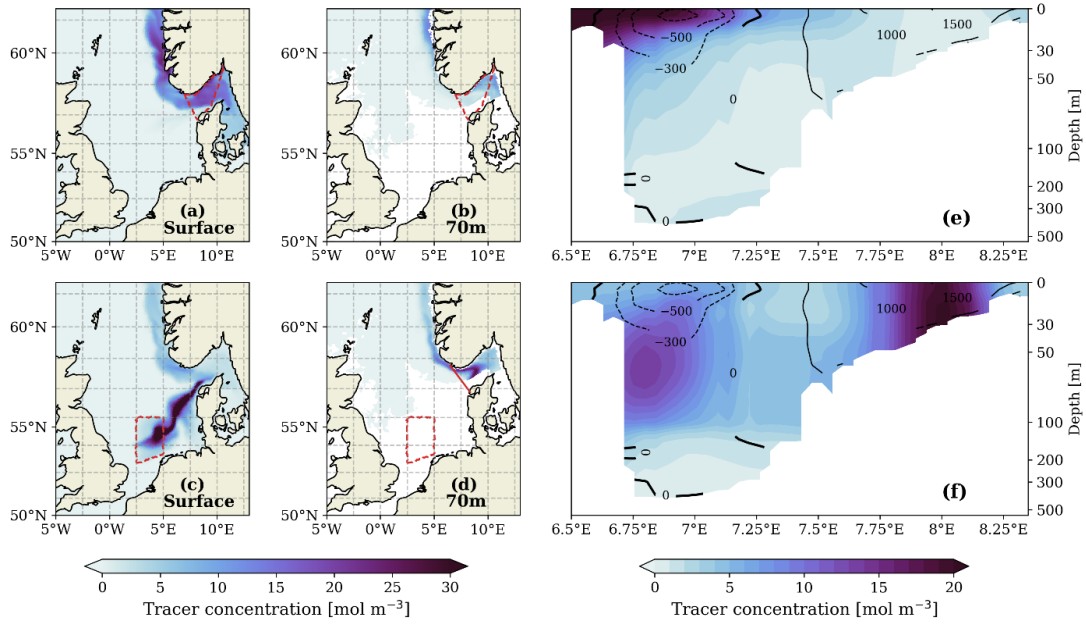

**Figure 5: Passive tracer distribution after 320 days of continuous injection in the Skagerrak (a-b and e) and in the southern North Sea (c-d,f). (a) and (c) show the tracer distribution at the surface while (b) and (d) show the distribution at the 70-meter depth. Red dashed lines highlight the areas of tracer injection from the surface. (e) and (f) show the vertical distribution of the tracer over the same transect as shown in Fig.4. The location of the transect is marked by the solid red line in Fig.4d.**





## 3.2 Enhanced oceanic $CO_2$ uptake

The OAE-induced oceanic $CO_2$ uptake occurs far beyond the deployment sites (Fig. 6). Areas with excess $CO_2$ uptake extend from the deployment sites to the Norwegian coast. The maximum increase in $CO_2$ flux is observed at the deployment sites,

with the magnitude varying among the three scenarios according to the rate of alkalinity addition (Table 1). The AE_ShipTrack scenario, which has the lowest maximum $CO_2$ uptake flux, is characterized by the largest spatial scale of detectable uptake, with an OAE-induced $CO_2$ uptake flux of 0.05 mol/m²/yr extending into the North Atlantic as far as 65°N (Fig.6c). This pattern cannot be explained by surface alkalinity availability alone (Figs. 3a-c), as the amount of added alkalinity in the upper mixed layer in the Norwegian Trench is similar across the three scenarios. This discrepancy suggests that $CO_2$ uptake potential

is influenced not only by alkalinity levels but also by the background carbonate conditions.

In all the three scenarios, oceanic carbon uptake reaches a new equilibrium within one year (Fig. 7a). Although the European coast area used for OAE is three times larger than the German EEZ OAE area, the total $CO_2$ uptake is similar for both implementations, reaching approximately 4.8 megaton of $CO_2$ removal per year. This is because the alkalinity is redistributed in a similar manner in both cases, resulting in comparable alkalinity availability for $CO_2$ sequestration (Fig. 2 and Fig.3a-b).

The AE_ShipTrack scenario achieves nearly 0.8 megatons less $CO_2$ uptake than the other two scenarios, mainly due to the smaller amount of alkalinity remaining at the surface (Fig. 2).

The $CO_2$ uptake efficiency ($\eta CO_2$), defined as the ratio of cumulative excess $CO_2$ uptake to the total amount of added alkalinity, approaches a stable level over time. After one year, $\eta CO_2$ ranges from 0.5 to 0.7 across the three scenarios, and after 5 years, it reaches a plateau ranging from 0.65 to 0.8 (Fig.7b), consistent with findings from previous studies (Burt et al., 2021; He and

Tyka, 2023). The AE_ShipTrack scenario shows a lower efficiency (~0.66) compared to AE_EUCoast and AE_GerEEZ (~0.79). Notably, due to the slow air-sea gas exchange, not all added alkalinity in surface fully equilibrates with the atmosphere in these continuous addition scenarios. Higher efficiency would likely be achieved with complete alkalinity utilization, as shown in an additional experiment (AE_EUCoast_Ter), where alkalinity addition in AE_EUCoast is stopped in 2008, allowing the model to run without OAE until 2010. In this case, $\eta CO_2$ slightly increases to ~0.8 (Fig.7b). Additionally, part of the $CO_2$

uptake potential is lost at the model boundaries, which cannot be quantified with the current model system.





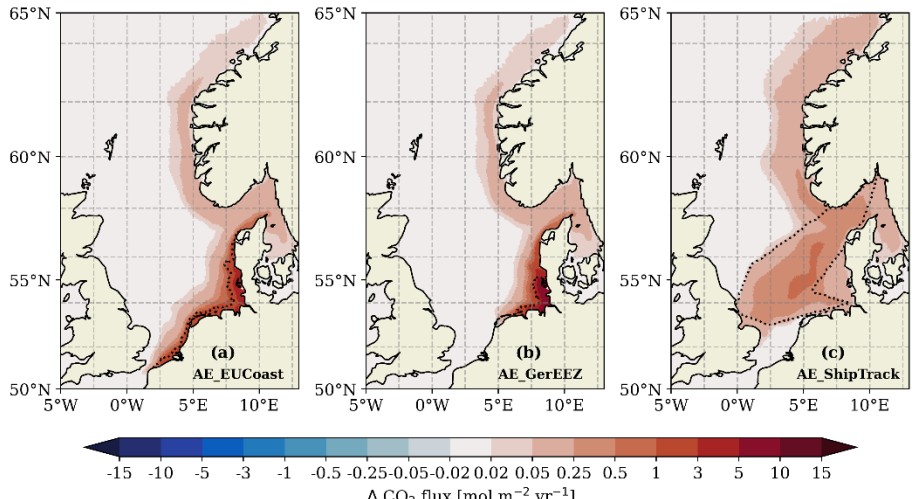

**Figure 6: The spatial distribution of excess $CO_2$ uptake averaged over 2002-2010 in the three OAE scenarios. Note the nonlinear color bar.**

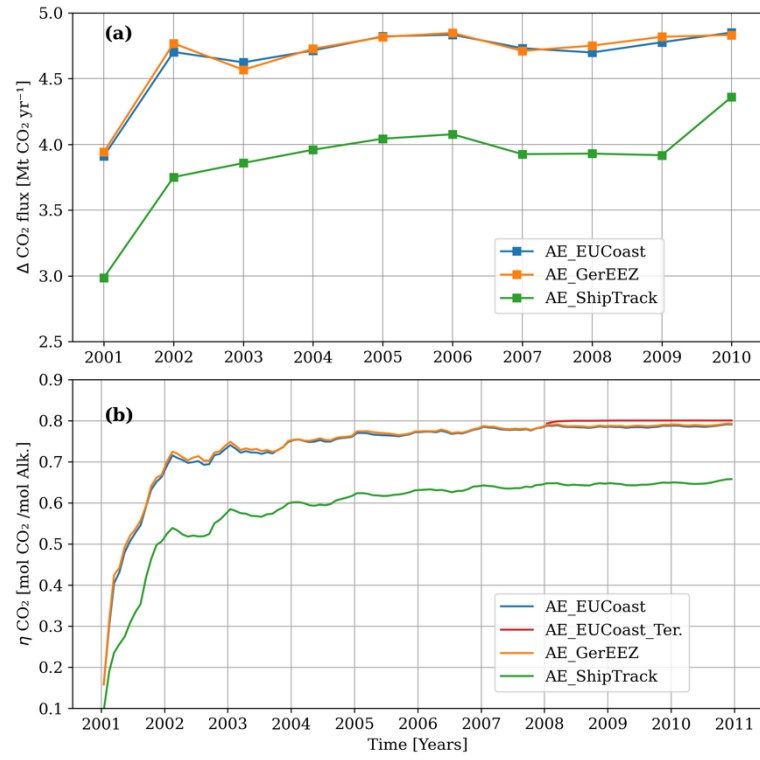


**Figure 7: (a) The yearly excess $CO_2$ uptake over the model domain due to OAE in different scenarios. (b) The $CO_2$ uptake efficiency ($\eta CO_2$) with a unit of mol $CO_2$ uptake per mol alkalinity addition, defined as the accumulated $CO_2$ uptake relative to the accumulated alkalinity addition.**



### 3.3 Enhanced carbon storage and cross-shelf export

In contrast to global models, tracers are lost at the open boundaries of the regional model. Despite this, we assess the enhancement of local carbon storage—namely the excess DIC retained within the model domain—as an initial measure of OAE efficiency. In all scenarios, two primary regions show significant DIC accumulation: one at the vicinity of the OAE deployment site and the other along the deep trench of the Norwegian coast (Figs. 8a-c). The latter acts as a major reservoir as nearly all excess DIC reaching below the permanent pycnocline accumulates here (Figs.8d-f). In the AE_ShipTrack scenario,

this sink area shows lower DIC accumulation throughout the water column compared to the other two scenarios (Fig.8c); however, below the permanent pycnocline, the DIC accumulation is comparable across all three scenarios (Fig.8f).

$CO_2$ absorbed by the ocean has the potential to re-enter the atmosphere through processes like surface warming, ocean circulation, and change of carbon chemistry. For effective CDR, it's essential that added carbon enters the ocean interior and stays isolated from atmospheric exchange over timescales relevant to carbon management. In this context, DIC below the

mixed layer serves as a primary indicator of carbon storage potential, as it has the chance to be transported into the deep ocean (Fig. 9a, dashed lines). In the AE_EUCoast and AE_GerEEZ scenarios, 42.7% and 42.5% of total excess DIC within the model domain resides below the mixed layer, while in the AE_ShipTrack scenario, this fraction increases to 50.8% (Fig. 9b). DIC at these depths may still be mixed back to the surface. Long-term storage potential is more accurately represented by DIC that reaches below the permanent pycnocline, where it is less likely to return to the surface (Fig. 9a, solid thin lines). In the

AE_EUCoast and AE_GerEEZ scenarios, this portion accounts for less than 5% of the total excess DIC stored in the model domain, whereas in the AE_ShipTrack scenario, nearly 9% of the total excess DIC reaches these depths, indicating a slightly higher storage capacity. Notably, the deep ocean DIC inventory does not further increase over time, suggesting an effective transport to regions beyond the model domain.




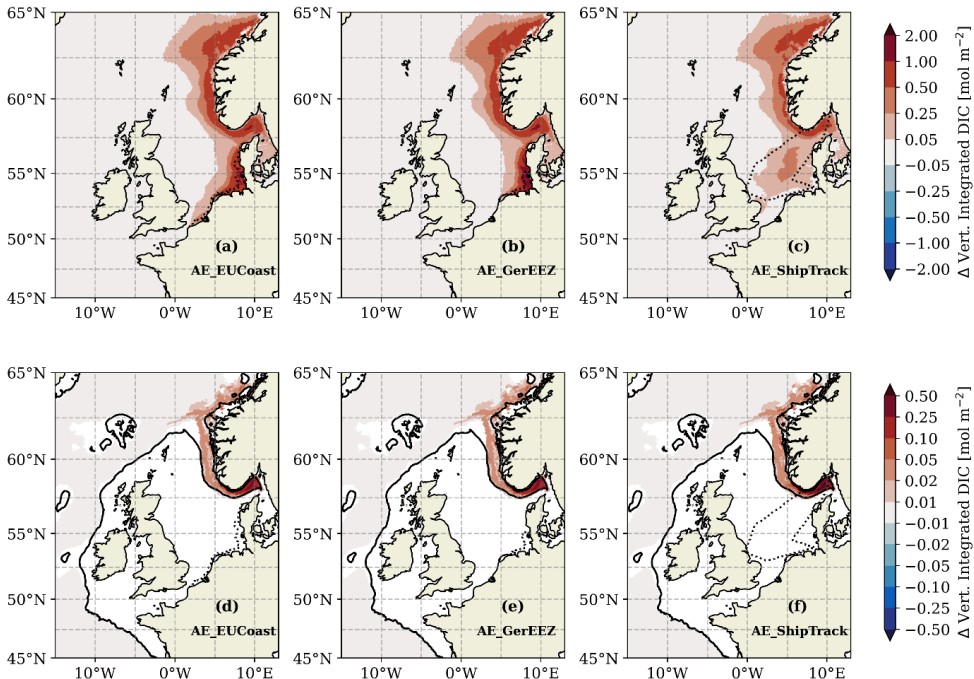

**Figure 8: Difference of vertically integrated DIC between the OAE scenarios and the CTL simulation over the whole water column (a-c) and below the permanent pycnocline (d-f) in the last simulation year (2010). Note the nonlinear color bar. The black lines in (d-f) represent the 200m isobath. In (d-f) white indicates areas where no permanent pycnocline is present.**

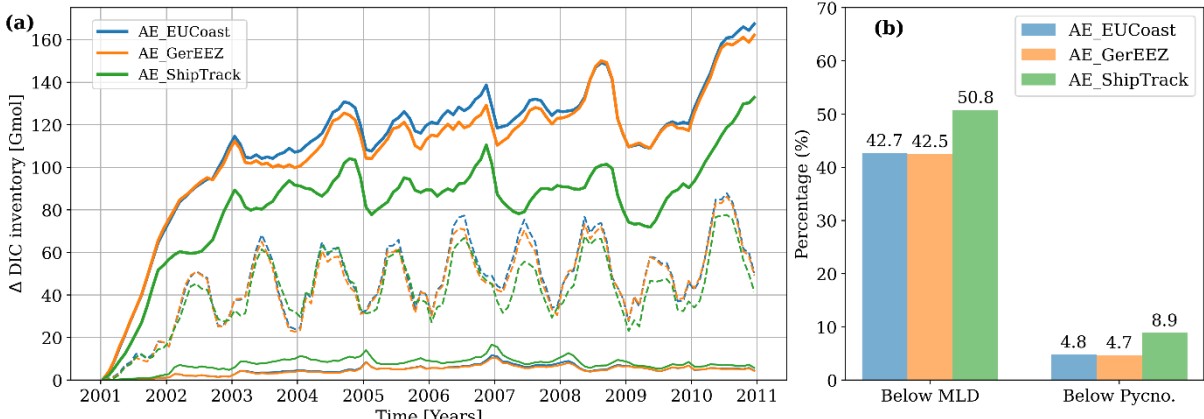

**Figure 9: (a) The excess DIC inventory across the entire model domain, integrated over the entire water column (solid thick lines), below the upper mixed layer (dashed lines) and below the permanent pycnocline (solid thin lines). (b) the percentage of the total excess DIC inventory below the mixed layer depth (Below MLD) and below the permanent pycnocline (Below Pycno.). The calculation is based on the model results of 2006-2010.**

It is therefore meaningful to quantify the increased transport of carbon from the North Sea to the deep North Atlantic Ocean as a second metric to assess the long-term OAE efficiency (Graham et al., 2018; Holt et al., 2009). This transport is normally



calculated as carbon export across the shelf-break, which is defined as the 200m isobath and is closed by two gates extending to the Norwegian and French coastline at the northern and southern limits of the region (Fig.1, and Graham et al. 2018). Figs.8d-f imply that the major increased transport occurs at the Norwegian Trench, as is also evidenced from the spatially resolved cross-shelf transport with respect to the CTL run (Fig.S3 in the Supplement S3).

The depth profile of DIC fluxes along the Norwegian Trench transect (the northern gate shown in Fig. 1) shows greater export
in the AE_EUCoast and AE_GerEEZ scenarios within the upper 80 meters, which is above the permanent pycnocline depth represented in our model (Fig. 10). In this region, our simulation captures a permanent pycnocline depth around 200 meters. Below this depth, enhanced DIC export is comparable across all three scenarios, with the AE_ShipTrack scenario showing a slightly larger value. The total excess DIC export below the permanent pycnocline ranges from 3.17 to 3.81 Gmol C/year, representing efficient deep ocean transport for long-term storage. This deep export accounts for only 3%, 2.9%, and 4.1% of
the excess atmospheric $CO_2$ uptake in the three scenarios, respectively.

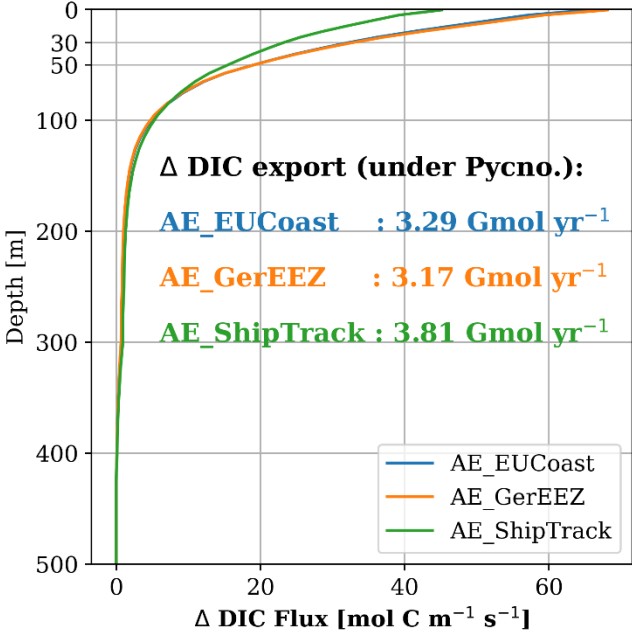

**Figure 10: Difference of the DIC flux per unit depth integrated over the transect in the Norwegian Trench (the northern gate shown in Fig.1) relative to the CTL run for the three scenarios. Positive values indicate DIC export to the Atlantic Ocean. The amounts of**
**excess DIC transported below the permanent pycnocline for each of the three scenarios are overlaid on the plot. The calculation is based on the model results of 2006-2010.**



## 3.4 Changes in carbonate chemistry

Increasing alkalinity alters the activities of all $CO_2$ species within the carbonate system (Middelburg et al., 2020; Zeebe and Wolf-Gladrow, 2001) and affects acid-based buffering capacity, both critical to marine organisms (Riebesell and Tortell,
2011). To examine these effects further, the OAE-induced changes in proton concentration ($H^+$) (thus pH) is quantified to present the shift in carbonate chemistry.

The three scenarios reveal varying degrees of $H^+$ change, with the largest shift observed in the AE_GerEEZ scenario and the smallest in the AE_ShipTrack scenario (Fig. 11). In all cases, the most pronounced changes occur directly at the deployment sites. The AE_EUCoast and AE_GerEEZ scenarios display concentrated areas of higher shifts, while the AE_ShipTrack
scenario shows a more diffuse pattern of change. Specifically, in the AE_EUCoast and AE_GerEEZ scenarios, $H^+$ changes are prominent along the European coast, with values highest nearshore and tapering off offshore. In the AE_GerEEZ scenario, the maximum decrease in $H^+$ concentration corresponds to an approximate increase in pH from 8.1 to 8.4 (Figs. 11b and 12). In contrast, the AE_ShipTrack scenario results in a change nearly an order of magnitude smaller than in the other scenarios (Figs. 11c and 12). Additionally, a pH change of about 0.01 from the baseline value of around 8.1 is detected in the Norwegian
Trench across all three scenarios (Fig. 11).

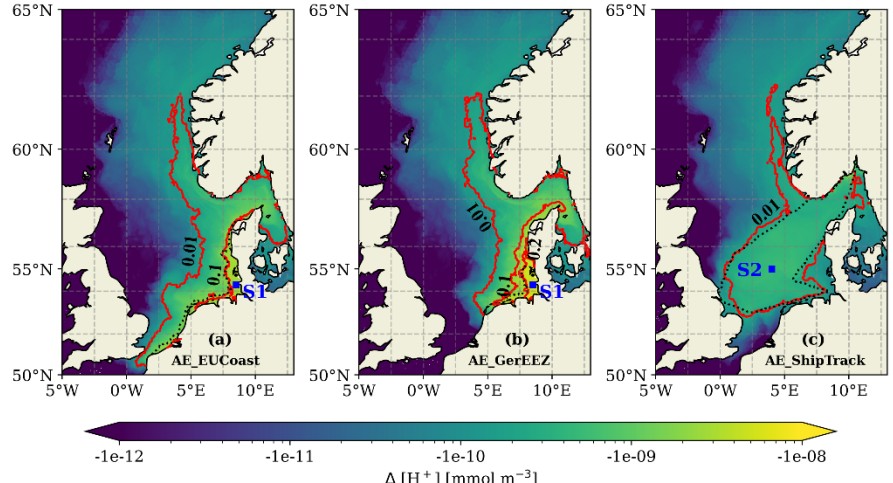

**Figure 11: The maximum changes in H⁺ concentration during the OAE periods for each scenario as shown by spatial colors. Overlaid red contour lines indicate the corresponding pH changes. Black dotted lines outline the OAE sites. Squares mark the stations where time series of H⁺ change and pH are plotted in Fig.12. Note the logarithmic scale of the color bar.**

To examine the temporal evolution of pH changes in each OAE implementation, we select one station per scenario where the $H^+$ concentration shows the greatest decrease. The AE_EUCoast and AE_GerEEZ scenarios share a station along the German coast, marked in Figs. 11(a) and 11(b), while the AE_ShipTrack scenario's station is located in the central North Sea (Fig. 11c). The daily time series of $H^+$ change relative to the CTL simulation at these stations is shown in Fig. 12(a). $H^+$





concentration changes display marked seasonality without a significant trend throughout the experiments. These changes do
not notably disrupt natural seasonal patterns (Figs. 12b–c). In the AE_GerEEZ scenario, pH levels occasionally exceed the
observed upper limit of 8.4, suggesting potential implications for the North Sea ecosystem (Carstensen and Duarte, 2019).

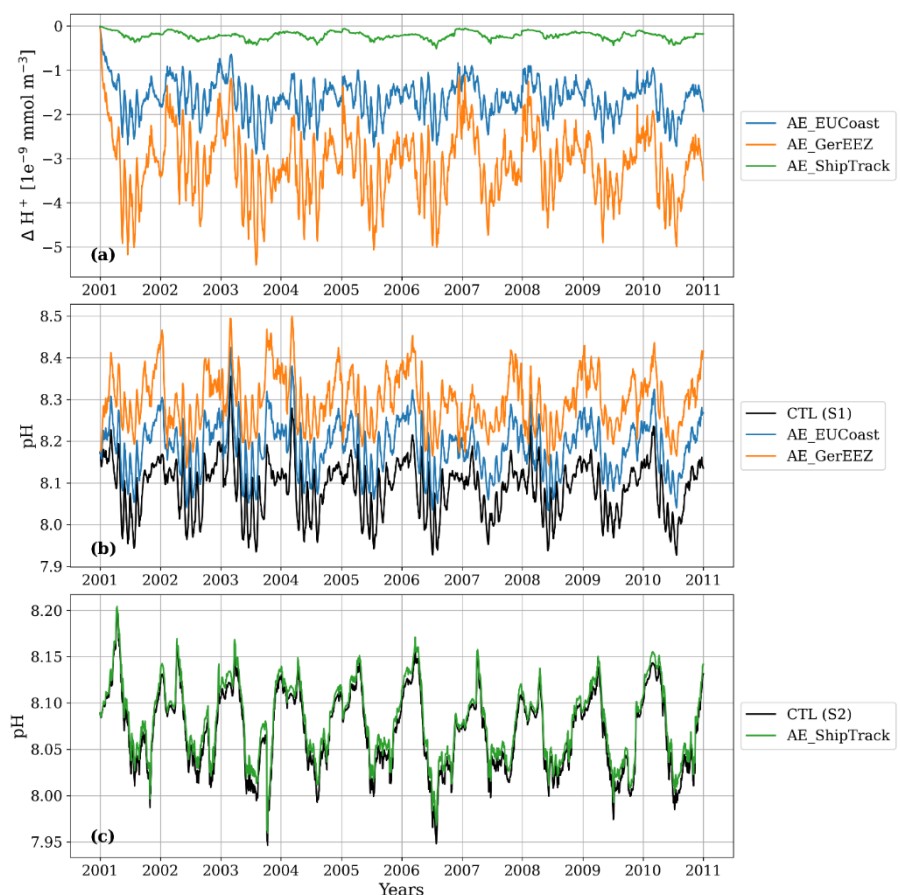

**Figure 12: (a) Time series of H⁺ changes resulting from the OAE implementations. For the AE_EUCoast and AE_GerEEZ scenarios,**
**the time series is taken at location S1, as marked in Figs. 11(a) and (b). For the AE_ShipTrack scenario, the time series is taken at**
**location S2, as marked in Fig. 11(c). (b) Time series of pH at location S1 with and without OAE perturbation. (c) Time series of pH**
**at location S2 with and without OAE perturbation.**



## 4 Discussion

### 4.1 General discussion

This study provides an initial assessment of OAE in the North Sea, with a focus on how physical factors like ocean circulation and vertical mixing influence OAE effectiveness. To achieve this, we employ idealized scenarios where DIC-free alkaline material (e.g., NaOH) is continuously introduced at the sea surface in selected locations, offering a clear framework for analysis. These simulations allow us to capture key outcomes, including the fate of added alkalinity, associated $CO_2$ uptake,

enhanced carbon storage and export, and changes in key carbonate chemistry parameters. However, due to the limitations of the regional model, our analysis is confined to the NWES region, and any potential OAE impacts outside this domain are not considered.

The $CO_2$ uptake from the atmosphere reaches a steady state within two years (Fig. 7a), which is faster than in most open seas (Burt et al., 2021) and enclosed coastal seas like the Bering Sea (Wang et al., 2023). This rapid response is attributed to the

relatively short flushing time of the North Sea (Blaas et al., 2001). These physical characteristics prevent the local accumulation of added alkalinity at the surface (Fig.2, dashed lines), thereby avoiding persistent changes in local carbonate chemistry (Fig.12). Simultaneously, the rapid water exchange between the North Sea and the Atlantic Ocean facilitates the export of absorbed $CO_2$, as revealed by Thomas et al. (2005), who found that over 90% of the atmospheric $CO_2$ taken up by the North Sea is transferred to the Atlantic Ocean. This dynamic also leads to the loss of alkalinity beyond the target domain, causing

$CO_2$ uptake to occur elsewhere, which is not accounted for by the current model. As a result, the $CO_2$ uptake efficiency estimated here is lower than what might be expected in reality. Burt et al. (2021) using a global model that accounts for all $CO_2$ exchanges, found that continuous alkalinity addition to selected regions achieves a $CO_2$ uptake efficiency ($\eta CO_2$) of 0.7–0.77, with exceptional higher efficiencies only in global (0.79–0.8) and Southern Ocean deployments (0.83–0.84). The 0.65-0.8 efficiency range estimated in our study (Fig.7b) suggests that OAE in the coastal North Sea could be more effective than

in much of the open ocean. Comparisons with coastal tests by He and Tyka (2023) reveal that the North Sea ranks among the most efficient coastal areas. However, our three scenarios underscore the critical role of injection site in determining OAE effectiveness. For instance, offshore injection, as in the AE_ShipTrack scenario, shows notably lower efficiency (Fig. 7). Injection frequency is another factor, as demonstrated by an additional scenarios where alkalinity is added only in the first month of a four-year simulation (Supplement S2 and Fig.S2). In those scenarios, despite that surface alkalinity has fully

equilibrated with the atmosphere, efficiency is lower than with continuous additions. Consequently, efficiency estimates remain highly variable, depending on the deployment methods such as injection site and frequency.

The basin-scale circulation of the North Sea is characterized by the only net outflow along the Norwegian coast via the Norwegian Coastal Current (Svendsen et al., 1991; Winther and Johannessen, 2006). This circulation pattern drives the robust transport of elevated alkalinity and DIC across all three scenarios—through the Skagerrak to the Norwegian Trench, and





ultimately exported to the North Atlantic Ocean—despite differences in deployment locations. The Norwegian Trench and Skagerrak have been identified as potential sinks for tracers from the North Sea (Dowdall and Lepland, 2012; Holt et al., 2009). However, their role in sustaining OAE effects is complex. While alkalinity is significantly lost to the deep, reducing the immediate impact of OAE, the concurrent transport of increased DIC to the ocean interior helps maintain the long-term effects of OAE. In practical applications, efforts should be made to minimize alkalinity loss in this region. Given their importance in

OAE processes, these areas are critical for CDR measurement, reporting, and verification (MRV).

Long-term carbon sequestration in OAE, which depends primarily on how much DIC is transferred and retained in the deep ocean (Legge et al., 2020), has received limited attention in past studies. Global model evaluations often approximate this storage using excess DIC inventory below a certain depth (e.g. the permanent pycnoclines) (Herzog et al., 2003). However, in regional models like the North Sea, open boundary transport complicates these estimates, as DIC can exit the model domain.

In such cases, long-term storage can be assessed through local deep storage and cross-shelf DIC export below the permanent pycnocline, where it is less likely to re-enter surface exchanges. Our model estimates this efficient export at around 2.9-4.1% of the total excess atmospheric $CO_2$ uptake, which aligns well with global estimates of 3.3-6.3% (Nagwekar et al., 2024). Interestingly, our model results suggest that higher $CO_2$ uptake does not necessarily translate to greater long-term carbon sequestration. This is demonstrated by the AE_ShipTrack scenario, where $CO_2$ uptake efficiency is lower than in coastal

implementations but results in slightly higher long-term storage.

**4.2 Limitations of the regional model approach**

As a simplified approach, our regional model exposes several limitations in considering the real-word OAE implementation. As already mentioned in Section 4.1, the main limitation is that the model cannot account for tracers that exit the model domain, leading to an underestimation of $CO_2$ uptake potential. Once alkalinity and DIC are transported beyond the model boundaries,

they can no longer be tracked. To overcome this limitation, future investigations should consider nesting the regional model within a global model to better capture these processes.

We assume that DIC-free alkaline material ($NaOH$) is added, meaning only alkalinity enters the system without any accompanying dissolved inorganic carbon. In real-world deployments, the type of alkaline material added to the ocean is a crucial issue that requires thorough discussion, as different materials offer varying levels of effectiveness (Ilyina et al., 2013).

The addition of $CaO$, $Ca(OH)_2$, or $NaOH$ has the same impact on alkalinity, with 1 mol of each increasing alkalinity by 2 mol, while DIC remains unchanged. In contrast, when using $CaCO_3$, adding 1 mol of $Ca^{2+}$ and 1 mol of $CO_3^{2-}$ increases alkalinity by 2 mol but also adds 1 mol of DIC, leading to lower effectiveness for $CO_2$ sequestration. The use of $Ca(HCO_3)_2$ results in even lower effectiveness, as adding 1 mol of $Ca(HCO_3)_2$ increases both alkalinity and DIC by 1 mol each, which might drive outgassing. Beyond the choice of mineral, lab results have shown that the method of alkalinity addition—e.g.

whether in solid form or as a solution, and whether that solution is pre-equilibrated with the atmosphere—is crucial for





maintaining elevated alkalinity levels and avoiding net loss through carbonate precipitation (Hartmann et al., 2023). Additionally, the selection of material should consider dissolution rates, environmental impacts, and the desired timescale for $CO_2$ removal. However, a detailed examination of the alkalinity addition methods falls outside the scope of this study.

Elevated alkalinity can drive alkalinity loss through solid carbonate precipitation (Hartmann et al., 2023; Moras et al., 2022; Morse et al., 2007), especially in areas close to the addition sites. Although the surface ocean is already supersaturated with respect to calcite and aragonite (Hartmann et al., 2023), the precise conditions under which spontaneous abiotic mineral precipitation occurs remain less constrained (Feng et al., 2017; Marion et al., 2009). In our scenarios, the maximum alkalinity increase is at around 500 µmol/kg (Fig.S4 in Supplement S3), staying still below the critical threshold of ~1000 µmol/kg for avoiding runaway carbonate formation (Suitner et al., 2024). Additionally, incubation experiments with beach sand have shown that anthropogenic alkalinity can significantly reduce the natural alkalinity generation by inhibiting calcium carbonate dissolution from the sand (Bach, 2024). These factors, which have no significant effects under the current scenario conditions, remain under investigation and are not addressed in this study.

Excessive alkalinity additions, along with the associated shifts in carbonate chemistry, could perturb ocean biogeochemistry and lead to undesirable ecological side effects (Bach et al., 2019; Ferderer et al., 2022; Subhas et al., 2022). While considerable research has focused on the effects of ocean acidification (low pH) on marine life, relatively few studies have examined the impact of elevated pH and alkalinity on marine ecosystems (Renforth and Henderson, 2017). It remains unclear whether increased alkalinity and pH would benefit or harm marine organisms. Laboratory experiments have found that high pH levels can reduce photosynthetic rates due to bio-available carbon limitation (Hansen, 2002; Pedersen and Hansen, 2003; Rasmussen et al., 1983). In contrast, microcosm studies have shown that more severe changes in carbonate chemistry under unequilibrated conditions did not have a significantly greater impact on phytoplankton and heterotrophic bacteria communities than equilibrated conditions (Ferderer et al., 2022). In this study, the largest changes in pH predicted by our simulations (0.3 with a baseline pH of 8.1) remain within the range of natural variability, suggesting no dramatic effects on the local ecosystem. We expect much larger shifts in carbonate chemistry near the site of alkalinity addition, as discussed in the next section. The ecological impact in those areas should be carefully evaluated in the future.

## 4.3 Sub-grid scale change in carbonate chemistry

In addition to assessing the perturbation of ocean parameters on the basin scale, it is crucial to examine the acute, local-scale impacts that occur close to the deployment site, where conditions are highly sensitive to the method of addition. This may not be a concern for gradually dissolving alkalinity sources, such as ground olivine (Hangx and Spiers, 2009), but is highly relevant for rapidly dissolving materials. Some key research questions, such as the near-field effects of alkalinity feedstocks or potential impacts from secondary $CaCO_3$ precipitation from a point source, require models capable of resolving small scale processes (e.g. turbulent motion) (Fennel et al., 2023). These impacts, which occur over short timescales, can temporarily push the local



carbonate system into an extremely alkaline state, causing dramatic environmental changes at local scales (Renforth and Henderson, 2017). This issue is likely to arise when scaling multiple single point-source interventions to a level substantial enough to produce a measurable impact over a large area.

To estimate the local impact of point-source alkalinity addition at a finer scale, we perform an analytical assessment of sub-grid scale processes. We assume that five submarine outfalls are operated in the German Wadden Sea to achieve the total alkalinity delivery of 134 Gmol per year. Each outfall discharges the alkaline solution continuously at a constant rate (R). Given that the Wadden Sea is a tidally dominated shallow water body, we further assume that the discharge pipelines are positioned on the offshore seabed to prevent dry-land discharges during low tide. In this simplified estimation, we focus on

the short-term spread of alkalinity over periods shorter than a typical half-tidal cycle (flood/ebb period), since seawaters of the North Sea and the Wadden Sea are exchanged within each tidal cycle (Laane et al., 2013).

During this time, mass transport is driven by tidal currents. We assume the discharged alkalinity disperses into a cylindrical area, with the discharge point located at the cylinder's bottom center (Schematic shown in Fig.S5 in Supplement S3). The cylinder's radius is determined by the tidal current velocity (U) and the time (t). Additionally, we assume that within the given

time period (t), the alkaline solution becomes fully dispersed within the cylinder, leading to a uniform increase in alkalinity concentration calculated as follows:

$$\Delta TA = \frac{Rt}{\pi (Ut)^2 D},$$ where D is the water depth.                                                   Eq.(5)

Givern a typical mean tidal current velocity of 0.4m/s in the German Wadden Sea (Hagen et al., 2022) and a mean water depth of 5m, the alkalinity change can be estimated as a function of time, as illustrated in Fig.13. Assuming a background alkalinity

of 2300 µmol/kg, DIC of 2100 µmol/kg, a temperature of 15°C, and a salinity of 32 psu, with an initial pH of 8, we further calculate the pH change using PyCO2SYS (Humphreys et al., 2022), as shown in Fig.13.

The estimation shows that changes in alkalinity and pH decrease exponentially over time as the alkalinity is gradually diluted across an expanding area. In the immediate vicinity of the addition site (~200m²), the increase in alkalinity is predicted to reach beyond 4,500 µmol/kg, significantly surpassing the critical threshold for alkalinity runaway (Hartmann et al., 2023;

Suitner et al., 2024). This dramatic rise in alkalinity causes a subsequent increase in pH from 8 to 11, which is an order of magnitude greater than what is estimated with the current model resolution.

According to Hartmann et al. (2023), alkalinity loss becomes noticeable with additions exceeding 1000 µmol/kg. This level of alkalinity increase corresponds to an affected area of approximately 0.07 km² per outfall application. In the Wadden Sea, the pH experiences a long-term change from 7.8 to up to 8.4 (Martens, 1989; Provoost et al., 2010; Rick et al., 2023). Under the

outfall scenario, the area with an increase of pH to 9 covers roughly 0.06 km², while the area affected by a pH increase to 8.5




expands to around 0.42 km². When scaled up to account for all five outfalls, the affected area with pH changes comparable to the local natural variations exceeds 2 km². Within these areas, both the ecosystem and natural chemical process are likely to be perturbed. The growth of the pelagic primary producers and the macrobenthos might be reduced and even stopped for certain species at high pH levels (Hansen, 2002; Pedersen and Hansen, 2003; Rasmussen et al., 1983). The pH change might

also cause species succession (Hansen 2002). Even though no direct studies in the Wadden Sea, it is speculated that the area might harbour more extensive blooms of calcifying phytoplankton (coccolithophores) and thick layers of calcareous ooze under the high alkaline conditions (Bach et al., 2019). While these sub-grid scale perturbations are not captured by the current model, thorough evaluation through OAE modeling across multiple spatial scales is essential.

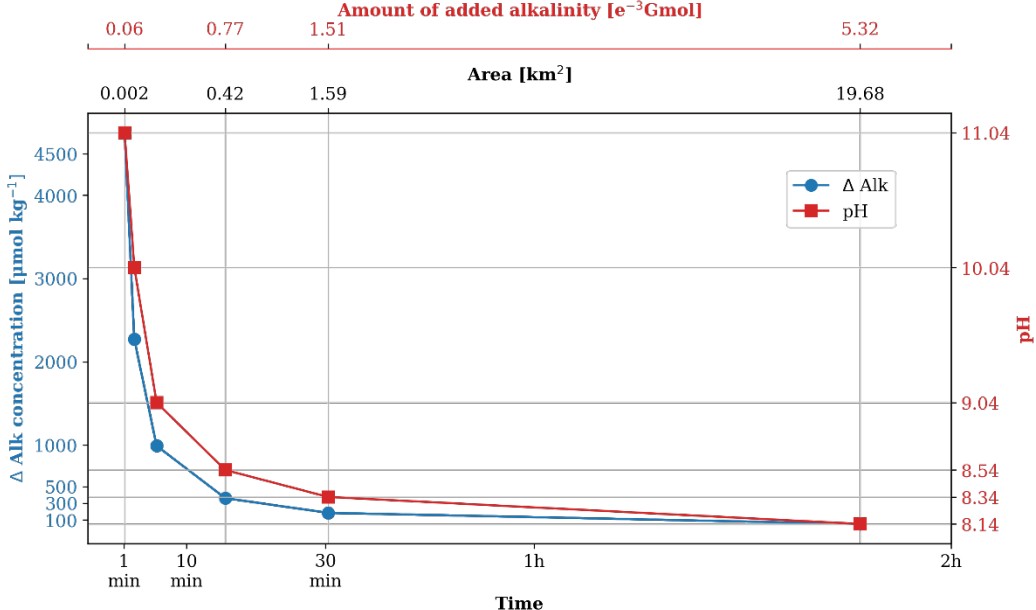

**Figure 13: Changes in alkalinity and pH over time in the ambient water surrounding a marine outfall that continuously discharges alkalinity solution into the ocean. The x-axis also indicates the corresponding size of the affected area and the amount of alkalinity added.**

## 5 Conclusions

An unstructured regional model is used in this study to simulate the addition of a fixed amount of alkalinity to the surface of

the North Sea, targeting the removal of 5 megatons of $CO_2$ per year. We select three areas likely to be suitable for OAE implementation based on their location and accessibility. The study examines the effects of OAE in terms of alkalinity redistribution, $CO_2$ uptake, DIC storage and cross-shelf transport, as well as potential side effects reflected in changes in carbon chemistry parameters such as pH.

Each OAE deployment location has its advantages and disadvantages. Spreading alkalinity over a larger coastal area (the European coast) and a smaller area (the German EEZ) result in similar OAE efficiency. However, concentrating alkalinity in the smaller German EEZ area significantly alters the carbon chemistry, resulting in a pH change more than twice as great as that observed with deployment along the European coast. While this approach reduces OAE costs, it also increases the risk to the local ecosystem.

Dispersing the material over a larger offshore area, such as the ship-covered area, leads to greater alkalinity loss to deeper waters. With less alkalinity available at the surface, $CO_2$ uptake is reduced. However, long-term DIC storage remains comparable to that of coastal deployments, resulting in similar DIC transport efficiency to the open ocean. In this scenario, the local water undergoes the least change in carbon chemistry, with alterations an order of magnitude smaller than that in the other two scenarios.

The regional model, with a maximum horizontal resolution of ~4.5 km, is insufficient to capture the local carbonate response on short time scales, leading to an underestimation of carbonate chemistry perturbations by more than an order of magnitude. Therefore, small-scale modeling approaches are needed to account for related processes in the nearfield of alkalinity additions.

*Data availability.* All data and code necessary to reproduce the results presented in this publication will be made available upon publication.

*Author contributions.* FL was responsible for the conceptualization of the study, while the methodology was developed collaboratively by FL, UD and CS. Software development was undertaken by JK, FL and KD. FL performed the formal analysis. The original draft of the manuscript was prepared by FL, and subsequent writing, review, and editing were contributed by FL, UD, JK, KD, HT and CS. FL also took responsibility for visualization.

*Competing interests.* The contact author has declared that none of the authors has any competing interests.

*Acknowledgments.* Computational resources for this work were made available by the German Climate Computing Center (DKRZ) through support from the German Federal Ministry of Education and Research (BMBF). This research was funded by the BMBF in the framework of RETAKE, one of the six research consortia of the German Marine Research Alliance (DAM) research mission "Marine carbon sinks in decarbonization pathways" (CDRmare), Grant 03F0895C.



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
