# Peer review of "Evaluating ocean alkalinity enhancement as a carbon dioxide removal strategy in the North Sea"

_EGUsphere, 2025_

## Author Comment (AC1)

**General response to Reviewer #1:** We sincerely appreciate the reviewer's recognition of our work and the thoughtful suggestions for improvement. In response to the main recommendation, we have added further details on the validation of the model simulation and the maximum attainable CDR efficiency in North Sea waters in the main text. In addition, we have carefully addressed all points raised. Please find below our detailed responses. The original text of the review is presented in **black,** while our responses are presented in **blue**.

**Reviewer#1' comments:**

Liu et al present a study of ocean alkalinity addition (OAE) in the North Sea using a regional ocean model. They vary the location of OAE deployment and explore the impacts on alkalinity transport, ocean carbon uptake and long-term carbon storage. This was an enjoyable read and is a really nice addition to the growing OAE literature that provides a number of interesting and often counterintuitive findings which are robustly explored. These findings include:

1. The North Sea is a relatively efficient OAE deployment region despite its shallow depth and short flushing time.

2. Small differences in the location of North Sea OAE deployment can lead to considerable differences in alkalinity dispersion, ocean carbon uptake enhancement and long-term carbon storage.

3. Loss of alkalinity to the deep ocean does not necessarily equate to inefficient CDR as although the total carbon flux enhancement is reduced, a greater proportion of this carbon may be transported below the permanent pycnocline.

The manuscript is clear and well-written and I recommend publication subject to a few minor corrections. My main suggestion is that the authors provide a little additional detail on the validation of simulated carbonate chemistry (even though this is published in greater detail elsewhere). Alongside this, it would be good to provide some detail on the maximum attainable CDR efficiency possible in North Sea waters based on mean present-day carbonate chemistry conditions.

Minor comments

1. L90 "efficiently transported" I recommend phrasing this differently.

**Answer:** Thank you for the kind suggestion. Indeed, "efficiently transported" might be unclear or overly broad. To improve the clarity, we have revised the sentence as follows:

"An optimal location would *facilitate the movement of seawater with elevated alkalinity, ensuring adequate atmospheric contact before it is transported into the ocean's interior,* thereby maximizing OAE efficiency while minimizing potential impacts on the local carbonate system and ecology."

The change has been made in the main text accordingly. See **lines 90-91.**

2. L101 Deployment areas appear stippled not "hatched".

**Answer:** Thank you for pointing out this mistake. 'hatched' has been changed to 'stippled'. See **line 102** in the main text.

3.  L161. I see that carbonate chemistry evaluation is published elsewhere. Nonetheless I think some comment on the ability of the model to simulate DIC/TA/pCO2 in the North Sea is required here.

**Answer:** Thanks for the reviewer's comments. Indeed, some comments on the ability of the model to simulate the key features of the North Sea carbonate chemistry is necessary to demonstrate whether the model is capable of being used in OAE studies. We therefore added a short summary of the model performance on simulating DIC/TA/pCO2/air-sea $CO_2$ flux in the main text. Please see **lines 195-202.**

It reads as: *A comprehensive validation of the model's physical dynamics, biological processes (e.g., primary production), and carbonate chemistry has been conducted by Kossack et al. (2023, 2024). The modelled DIC and TA exhibit quantitative agreement with observational data, although both simulated variables exhibit a slight overestimation. The model effectively captures the seasonality and spatial variability of surface $pCO_2$ and air-sea $CO_2$ fluxes on the NWES, despite some discrepancies between simulations and observations due to inherent model limitations. Nevertheless, the SCHISM-ECOSMO-$CO_2$ model system successfully reproduces key features of the marine carbon cycle on the NWES, both qualitatively and with reasonable quantitative skill (Kossack et al., 2024). Therefore, the model serves as a reliable tool for OAE studies.*

4.  L176. Can you clarify here whether there are sediment fluxes of alk and DIC and whether these fluxes are influenced by benthic carbonate chemistry and OAE?

**Answer:** Yes, our model includes sediment fluxes of alkalinity and DIC arising from sedimentary processes such as remineralization, nitrification, and denitrification. However, benthic carbonate chemistry is not explicitly represented in the current model. Those sediment fluxes are thus not directly influenced by benthic carbonate chemistry or OAE.

There are indeed studies suggesting that OAE could influence the natural production of alkalinity, particularly through sedimentary processes. However, these feedback mechanisms remain poorly understood, as only a few studies have explored this topic. At present, incorporating such processes into the model remains challenging due to the limited knowledge. Currently, we are actively working on this aspect by integrating laboratory and field findings into our model developments. This work is still undergoing and is not presented in the current manuscript.

To improve clarity, we have added a relevant description in the section of 2.1 model description in the main text, see **lines 142-143 and lines 149-151.**

5.  L301. Typo "of the" tracer

**Answer:** Thank you very much for pointing this out. We have corrected the typo, ensuring that the phrase now correctly reads 'of the tracer' in the revised manuscript, see **line 307.**

6.  L318-340 Some context on the maximum attainable uptake efficiency in the North Sea would be useful here. Is this around 0.85 given the mean simulated surface ocean DIC and alkalinity fields?

**Answer:** Given the mean simulated surface ocean DIC and alkalinity, we estimate the maximum attainable $CO_2$ uptake efficiency in the North Sea is around 0.86. The map is provided below and added to the supplement as **Fig. S6.**

A comparison of the model results with the maximum attainable $CO_2$ uptake efficiency as well as the related discussion is added to the main text, reading as '*However, this value remains below the maximum*

*attainable CO₂ uptake efficiency in the North Sea, which is determined by background alkalinity and DIC levels and reaches approximately 0.85-0.9 over the North Sea (Fig.S6 in Supplement S3). The lower efficiency observed in the scenarios primarily results from alkalinity loss to the subsurface layer and horizontal transport beyond the model boundaries.'*, see **lines 345-349.**

[Figure]

**Figure S6. Map of the maximum attainable CO2 uptake efficiency in the North Sea determined by the mean simulated surface ocean DIC and alkalinity fields.**

7.  Figure 9. It's difficult to distinguish between the thick and thin lines in panel a. I suggest using another line type (e.g. dotted or a different dash type).

**Answer:** We appreciate the reviewer's recommendation. Using a different line type indeed improves the clarity of the figure. In response, we have changed the thin solid lines in Figure 9a to dashed lines, making them more distinguishable from the thick solid lines. This adjustment enhances readability and makes it easier for readers to differentiate between the two types of DIC inventory. **Figure 9** is displayed below and has been updated in the manuscript, with **the caption of the figure** being changed accordingly.

[Figure]

Figure 9: (a) The excess DIC inventory across the entire model domain, integrated over the entire water column (solid lines), below the upper mixed layer (dashed lines) and below the permanent pycnocline (***dotted lines***). (b) the percentage of the total excess DIC inventory below the mixed layer depth (Below MLD) and below the permanent pycnocline (Below Pycno.). The calculation is based on the model results of 2006-2010.

8. L386. Recommend not using "captures"

**Answer:** Thanks very much for the suggestion. We have revised the sentence and replaced 'captures' with 'represents', which more precisely describes the role of the model in representing the pycnocline depth. The revised sentence reads: *"In this region, our simulation **represents** a permanent pycnocline depth around 200 meters."* See **line 396.**

9. L389-390. Are these values still increasing at the end of the simulations? Does this explain why the deep DIC inventory plateaus within the model domain?

**Answer:** No, the total excess DIC export below the permanent pycnocline stabilizes from 2003, after which there is no significant increasing trend but only interannual variability. This is evidenced by the yearly export illustrated in Fig. S7, which we have added to the supplement. The information is added in the main text as well, see **lines 398-399.**

This explains why the deep DIC inventory plateaus within the model domain.

[Figure]

**Figure S7. The yearly total excess DIC export below the permanent pycnocline across the Norwegian Trench for the three scenarios.**

10. L454-456. Can the authors explain this finding here as they do in the supplementary material? How do they know full equilibration has occurred? I recommend phasing this as CO2 system equilibration not alkalinity equilibration.

**Answer:** Yes, we added a more detailed explanation about how the scenarios are designed and what the results imply to the main text as follows (see **lines 463-470**): '*Injection frequency is another key factor, as demonstrated by three one-time alkalinity addition experiments. In these experiments, a fixed amount of alkalinity is added to the three selected locations only during the first month of a four-year simulation (Supplement S2 and Fig. S2). The extended simulation allows the seawater $CO_2$ system to gradually re-equilibrate with the atmosphere after alkalinity injection. This is evidenced by the $CO_2$ uptake efficiency plateauing at 0.57–0.76 mol $CO_2$ per mol of alkalinity after one year's simulation, indicating that no additional $CO_2$ uptake occurs beyond this timeframe. However, the $CO_2$ uptake efficiency is lower than in*

*scenarios with continuous alkalinity additions. A possible explanation is that in the one-time addition scenarios, alkalinity is introduced during winter, when strong flushing reduces residence time and thus leads to less effective equilibration.'*

As mentioned above, the $CO_2$ uptake efficiency stabilizes after one year of simulation, indicating that no further $CO_2$ uptake occurs beyond this point. This suggests that full equilibration has been achieved.

Thanks for correcting the expression regarding the CO2 system equilibration. Indeed, alkalinity itself does not equilibrate but rather the seawater (or its CO2 system) that interacts with the atmosphere. We have thoroughly checked the entire manuscript and the supplementary materials to ensure all related expressions have been corrected accordingly, see **lines 267,342-343** in the manuscript and **Supplement 2.**

In addition, we found that the buffer capacity in the North Sea is relatively homogeneous (Fig. S6) and therefore does not explain the low $CO_2$ uptake efficiency observed in the one-time alkalinity addition experiments. As a result, we have removed the sentence from the supplement: 'Another possibility is that with continuous alkalinity supply, the added alkalinity disperses over a larger area, where the water has a lower buffer capacity.' This deletion does not impact the overall content of the manuscript.

---

## Author Comment (AC2)

**Reviewer#2' comments:**

A clear and well-reasoned regional model study of ocean alkalinty enhancement in the North Sea. The study focuses on 3 "likely" hypothetical application scenarios with regards to potential EU or national implementations, which have largely been absent from previous global simulations.

This reviewer has 2 questions that they would hope the authors will consider and a very minor comment.

**Response:** We sincerely thank the reviewer for the constructive and insightful suggestions, which have greatly contributed to improving the paper. We have carefully addressed all points raised. Please find below our detailed responses. The original text of the review is presented in **black,** while our responses are presented in **blue**.

1. Given that the study is already constrained to the North Sea region, uses a model which has been published at finer scale and there exist global Earth System Model simulations at 1 km scale; why did the authors choose to perform simulations at ~4.5 km scales? The reviewer appreciates that the authors do discuss the necessity of further finer scale studies in future.

**Answer:** Indeed, high-resolution global Earth System Models (e.g., ICON (Adamidis et al., 2025)) and regional models (e.g. SCHISM-ECOSMO-CO2 (Kossack et al., 2024)) exist. However, we chose a setup at ~4.5 km resolution for the following reasons:

1) Computational efficiency:
   High-resolution models require immense computational resources. For instance, the high-resolution version (20 km–500 m) of SCHISM-ECOSMO-CO$_2$**,** which covers the same domain as our current model, has approximately **386,205 grid nodes and 758,387 triangular grid elements**. Running a one-year simulation with this setup requires about **one day on 2,176 CPUs**. In contrast, our configuration reduces the grid size to **71,158 nodes and 134,753 elements**, cutting the computational cost significantly—**a one-year simulation now takes only 18 hours on 1,024 CPUs**. This balance allows us to conduct long-term and multiple scenario simulations efficiently.
2) Sufficient resolution for regional-scale processes:
   The focus of our study is on the impact of alkalinity addition beyond the immediate application sites**,** where regional-scale processes (e.g., ocean circulation, vertical mixing) predominantly shape the system's response (see main text, lines -). Our chosen resolution is sufficient to resolve these key processes accurately, and the results remain **consistent with those from the higher-resolution version**, which has been validated and published (Kossack et al., 2024). To clarify this, we have added a brief summary of the model's performance in simulating DIC, TA, pCO$_2$, and air-sea CO$_2$ fluxes in the main text, see **lines 195–202.**

Considering both **computational feasibility and model performance**, we opted for this resolution as a well-balanced approach.

The necessity of finer-scale studies is explicitly discussed in **Section 4.3: "Sub-grid Scale Changes in Carbonate Chemistry,"** where we conclude:*"While these sub-grid scale perturbations are not captured by the current model, thorough evaluation through OAE modeling across multiple spatial scales is essential."* (lines ) This point is further emphasized in the **abstract**, which states: *"The model's resolution (~4.5 km in coastal areas) limits its ability to capture rapid, localized carbonate responses, leading to a nearly tenfold underestimation of chemical perturbations. Thus, finer-scale models are needed to*

*accurately assess near-source alkalinity impacts."* These discussions highlight the importance of future high-resolution studies to better resolve localized carbonate chemistry dynamics.

References:

Adamidis, P., Pfister, E., Bockelmann, H., Zobel, D., Beismann, J. O., and Jacob, M.:The real challenges for climate and weather modelling on its way to sustained exascale performance: a case study using ICON (v2. 6.6). Geoscientific Model Development, 18(4), 905-919, https://doi.org/10.5194/gmd-18-905-2025, 2025.

Kossack, J., Mathis, M., Daewel, U., Liu, F., Demir, K. T., Thomas, H., and Schrum, C.: Tidal impacts on air-sea CO2 exchange on the North-West European shelf. Frontiers in Marine Science, 11, 1406896, https://doi.org/10.3389/fmars.2024.1406896, 2024.

2. Could the authors please clarify Figure 13? This reviewer is unclear how the plotted lines are represented in axes of Time, Area (km2) and Amount of added alkalinity (e-3 Gmol) at the same time.

**Answer:** In this simplified estimation, we assume that:

1. Each outfall continuously discharges the alkaline solution at a constant rate (R).
2. The discharged alkalinity disperses through the mean tidal current into a cylindrical area, with the discharge point located at the bottom center of this cylinder (schematic shown in Fig. S5 in Supplement S3).

As a result, the radius (and consequently the area of the bottom side) of the cylinder is determined by the mean tidal current velocity (U) and the time elapsed since the start of alkalinity discharge (t). Meanwhile, the total discharged alkalinity is a function of the discharge rate (R) and time (t). **Therefore, both the affected area (the bottom side of the cylinder) and the total discharged alkalinity are functions of time (t) and can be presented along the same a-axis.**

To improve the clarity of the description regarding Figure 13, we have revised the relevant section of the main text, see **lines 546-553.**

It is important to note that the a-axis is non-linear, meaning that Time, Area, and the Amount of added alkalinity are all illustrated in a non-linear manner. We have also added a notation in **the caption of Figure 13** to clarify this. See **line 579.**

[Figure]

**Figure S5: Schematic showing how the alkalinity is discharged through a marine outfall.**

This reviewer wouold also like to suggest to the authors, with reference to line 11, that the effects of OAE are dependent upon the physical and biogeochemical conditions of a system.

**Answer:** Thank very much for the suggestion. We have improved the sentence by adding the 'biogeochemical' term. Now the sentence reads as '*The effect of OAE depends significantly on local physical and biogeochemical conditions*', **see line 11** in the main text.